# Translation control by altered start codon usage as a means of modulating the general stress response and virulence in *Listeria monocytogenes*

Jialun Wu[1*], Claire Kelly[1,2], Duarte N. Guerreiro[3], Brenda Chanza[1], Ashley Reade[1], Catherine M. Burgess[2], Conor O'Byrne[1*]

1 Bacterial Stress Response Group, Microbiology, Ryan Institute, School of Biological and Chemical Sciences, University of Galway, Galway, Ireland, 2 Teagasc Food Research Centre, Ashtown, Dublin, Ireland, 3 Pathogen Biology and Global Health, Centre for Ecology, Evolution and Environmental Changes (CE3C) and CHANGE - Global Change and Sustainability Institute, Faculdade de Ciências da Universidade de Lisboa, Lisboa, Portugal

* jialun.wu@universityofgalway.ie (JW); conor.obyrne@universityofgalway.ie (CO)

**Editor:** Danielle A. Garsin, University of Texas McGovern Medical School: The University of Texas Health Science Center at Houston John P and Katherine G McGovern Medical School, UNITED STATES OF AMERICA

## Abstract

In the food-borne pathogen *Listeria monocytogenes*, SigB is the central regulator of general stress response (GSR) and it mediates host entry by promoting acid resistance and epithelial cell attachment. However, mutations can readily arise to disable regulators of SigB (Rsb proteins), which suggests a considerable genetic plasticity in the GSR. To further investigate this, we defined the complete genome sequence of a clinical isolate and elucidated how sequential mutations within *sigB* operon (*rsbX* N77K and *rsbU* Q317*) impacted fitness through modulation of SigB activity. To investigate the plasticity of the GSR, we followed its genetic adaptation to lethal acidic challenge (mimicking the selective pressure encountered during entry into the host). Acid resistance developed rapidly and all 6 acid resistant derivatives (ARDs) selected for analysis had acquired mutations in *rsbW*, which encodes an antagonist of SigB that suppresses SigB activity during non-stress conditions. These mutations resulted in non-canonical start codons (*rsbW*ATG to *rsbW*ATA or *rsbW*ATT) or premature translation termination (*rsbW'*) and all were found to result in increased SigB activity. A translational reporter assay demonstrated distinct differences in translation efficiency between three start codons: ATG>ATA>ATT, suggesting that a perturbation of RsbW:SigB stoichiometry alters SigB activity. We then analysed start codon usage for all conserved genes in 60,692 *L. monocytogenes* genomes. This analysis revealed flexible usage of start codons associated with genetic clades in 39 conserved genes, 13 of which are involved in virulence and stress response. Further, we show that flexible use of canonical start codons (ATG and GTG) can also mediate different levels of expression of virulence and stress response genes. Taken together, we show the genetic plasticity of GSR regulation in a model pathogen, and highlight

**Data availability statement:** Raw data can be found at https://doi.org/10.5281/zeno-do.19196202. All other relevant data are in the manuscript and its Supporting information files.

**Funding:** This study was supported by Science Foundation Ireland Frontiers for the Future Programme (21/FFP-P/10 078) awarded to CO'B and the Irish Department of Agriculture, Food and the Marine (Project #2023RP994) awarded to CMB. J.W. receives salary from Science Foundation Ireland Frontiers for the Future Programme (21/FFP-P/10 078) as postdoctoral researcher, and C.K receives salary from the Irish Department of Agriculture, Food and the Marine (Project #2023RP994) as PhD student. The funders had no role in study design, data collection and analysis, decision to publish, or preparation of the manuscript.

**Competing interests:** The authors have declared that no competing interests exist.

the importance of translational control as a means of fine-tuning gene expression during short-term adaptation and long-term evolution for optimal fitness.

## Author summary

The general stress response (GSR) in foodborne pathogen *Listeria monocytogenes* is important for environmental stress response and for host entry, but GSR is also highly variable across wild isolates. In this study, we analysed the evolutionary trajectory of a clinical isolate with attenuated GSR and characterized the adaption by this strain to a host-mimicking stress (acidic condition). Under extreme selective pressure, mutations disabling a negative regulator of the GSR were enriched. Interestingly, several independently occurring mutations negatively affect the translation initiation by using non-canonical start codons. Prompted by this, we analysed the population-wide start codon usage by examining all available *L. monocytogenes* genomes (n = 60,690). This analysis revealed differential start codon usage in 39 conserved genes that are associated with different genetic clades. Furthermore, we demonstrated differential translation efficiencies between the different canonical start codons. This work highlights the genetic plasticity of GSR in the important food-borne pathogen *L. monocytogenes* and shows that altered translational start codon can be used as means of regulatory control. Together the data suggest that genetic changes in the regulation of the GSR might confer niche-specific fitness advantages.

## Introduction

The ethological agent of listeriosis, *Listeria monocytogenes* is a bacterial pathogen well-known to be capable of adapting to both saprophytic and pathogenic lifestyles. As a food and environmental microbe, *L. monocytogenes* is widely distributed and it is relatively resilient to low pH, low water activity, and low temperature [1]. Upon ingestion by humans, it can survive the stresses encountered in the gastro intestinal (GI) tract and cross the intestinal barrier, leading to systemic infection [2]. Symptomatic infections are typically associated with high mortality rates (estimated 20–30%) [3]. Despite the versatility of this bacterium, it has a relatively small (~3 Mbp) and conserved core genome [4,5]. Most *L. monocytogenes* isolates are from one of two phylogenetic lineages, I and II. Lineage I strains are more often associated with infection while lineage II strains are more often isolated from food and food processing environments [5]. In particular, several clonal complexes (CC) from lineage I are classified as hypervirulent (e.g., CC1, CC2, CC4, CC6) clades because of their frequent association with clinical cases and more severe disease outcomes (e.g., meningitis) [6].

The alternative sigma factor sigma B (SigB) is a very well conserved stress response and virulence regulator in *L. monocytogenes* [7]. Its activity is tightly controlled by a group of proteins encoded in a same operon (*rsbRSTUVW-sigB-rsbX*).

SigB is sequestered by the anti-sigma factor RsbW in unstressed conditions, which prevents SigB from associating with RNA polymerase and thus suppresses expression of the general stress response (GSR). Stress stimuli perturb the association of RsbT with a macromolecular complex known as the stressosome (which consists of RsbR and RsbS at a ratio of 2:1) [8]. The release of RsbT from the stressosome facilitates its interaction with the phosphatase RsbU [9]. RsbU then dephosphorylates the anti-anti sigma factor RsbV, which promotes its sequestration of RsbW, thereby releasing SigB [10,11]. Thus the partner switching that occurs between RsbW and RsbV and SigB in response to changes in environmental stress determines the activity of SigB and the expression of the GSR regulon. The last gene in the *sigB* operon, *rsbX* encodes a negatively acting regulator of SigB, which resets the sensing-ready state of the stressosome by dephosphorylation, allowing resequestration of RsbT [12]. Upon activation, SigB controls the expression of ~300 genes, approximately 10% of the entire genome content [13]. This global gene expression profile results in increased resistance to an array of environmental stresses including acid, salt, light and bile [7].

SigB also plays an essential role in host-cell entry. During passage through the acidic gastric fluid, the SigB-dependent glutamate decarboxylase and arginine/agmatine deiminase are important for the survival of *L. monocytogenes* [14,15]. Additionally, in the intestine, SigB controls the expression of a surface adhesin called Internalin A (InlA), which mediates epithelial attachment through an interaction with E-cadherin [16]. Although SigB activity in a lineage II laboratory strain was shown to be dispensable for systemic infection in a guinea pig infection model [17], recent evidence demonstrates that hypervirulent CCs or lineages are generally associated with high SigB activity [18]. These data suggest that the importance of SigB in the pathogenic lifestyle of this species has previously been underestimated. In line with this are the high levels of conservation on the *sigB* operon [19]. Apparently in contradiction with these findings is the observation from several groups that mutations attenuating SigB activity arise readily both *in vitro* and *in vivo* in both laboratory strains and wild isolates [18,20,21]. Indeed, we have previously demonstrated that loss-of-function SigB mutations confer fitness advantages under mild-stress conditions, although they are detrimental for surviving lethal challenges [22,23]. Hence, we hypothesise that SigB functionality is evolutionarily conserved in the long-term, while SigB activity can be fine-tuned through genetic adaptation to confer short-term conditional fitness advantages in specific environmental conditions.

To investigate this hypothesis the present study focused on a strain of *L. monocytogenes* CC1 (MQ140025) that was isolated from an ear swab in 2014 [24]. We previously reported that this strain displayed reduced acid survival and impaired SigB activity, likely due to lesions in the *sigB* operon (*rsbU* Q317* and *rsbX* N77K) [21]. Given the high level of conservation of RsbX N77, it was proposed that N77K negatively affects RsbX function and results in elevated activation of SigB. While RsbU Q317* abolishes a C-terminal $Mn^{2+}$ binding site of RsbU and likely renders RsbU inactive [25]. Strain MQ140025 presents as a *sigB*- strain likely because RsbU functions downstream of RsbX in the SigB activation pathway. The fact that SigB is regulated by a multi-step signal transduction pathway, means that there are multiple genetic targets (at least 7) that can be mutated to produce changes in its activity.

This study sought to investigate the plasticity of GSR, by adapting strain MQ140025 to an environment encountered within the gastric fluid of a host (acid stress challenge). We defined the complete genome of strain MQ140025 and subjected it to short-term adaption to a lethal acid challenge. The resistance mechanisms that conferred the stress adaptation were investigated. Arising from this analysis we exploited the NCBI database of *L. monocytogenes* genome sequences to investigate the evolution of start codon (SC) usage in conserved genes. The work presented reveals the genomic plasticity of the GSR in an intracellular pathogen and highlights altered translational initiation as a mechanism for fine-tuning gene expression during evolutionary adaption to environmental and host-related stresses.

## Results

### Clinical isolate shows rapid genetic adaption to lethal acidic stress

To test whether clinical strain MQ140025 (*rsbU* Q317*, *rsbX* N77K) could regain the SigB-mediated gastric acid survival, we designed an *in vitro* evolution experiment to select acid resistant mutants (Fig 1A). Cultures of strain MQ140025 were

exposed to repeated cycles of acidic stress (pH 3.0) followed by growth for the first two days and the same procedure with more acidic stress (pH 2.5) for the following two days (Fig 1B). A sample from each time point of the acid challenge was used as an inoculum to seed a tube of unacidified brain heart infusion (BHI) broth (1:2500). After recovery and incubation for 1 d at 37°C, the culture inoculated from the latest acid exposure time-point that successfully grew to stationary phase was re-exposed to acid challenge. An aliquot of each one of these cultures was also preserved as a -80ºC permanent stock. After the first two cycles of acid challenge (pH 3.0) and growth, acid resistance rapidly developed (Fig 1B) and therefore the recovered cultures were exposed to both pH 2.5 and pH 3 during the third acid challenge to ensure a high selective pressure. On day 3, the pH 2.5 treatment (Fig 1B, cycle 3') for 180 min resulted in no detectable survival, while the pH 3.0 treatment (Fig 1B, cycle 3) no longer fully inactivated the evolved cultures. Therefore, the cultures recovered from the day 3 (120 min at pH 2.5) treatment were re-exposed to pH 2.5 on day 4 (the last acid challenge). The development of acid resistance occurred reproducibly across three independent experiments (identical between parallel cultures, Fig 1B). These results show that acid resistance can be rapidly developed and selected in an acid sensitive strain of *L. monocytogenes*.

To quantify the development of acid resistance in 2-day (D2) and 4-day (D4) adapted cultures from all three independently evolved cultures (designated E1, E2, and E3), they were challenged with either pH 3.0 (Fig 1C) or pH 2.5 (Fig 1D) in comparison with F2365, a reference strain that belongs to the same clonal complex (CC1) and sequence type (ST1), and the parental strain MQ140025 (Table 1 and Fig 1C and 1D). Results confirmed the acid sensitive phenotype of MQ140025 while all acid adapted cultures survived better than MQ140025, albeit not as well as F2365. All D4 cultures were more acid resistant than D2 cultures. To investigate these differences in acid resistance, for each evolved culture that was tested in pH 2.5 survival experiments (Fig 1D) one colony from the latest surviving time point was re-streaked to prepare permanent stocks (t=90 min for D2 cultures; t=120 min for D4 cultures). In total this yielded 18 acid resistant derivatives (ARDs) from three independent pH 2.5 survival experiments (1st experiment: ARD1–6; 2nd experiment: ARD7–12; 3rd experiment: ARD13–18; Fig 1E). All 18 ARDs displayed a comparable level of acid survival (Fig 1E, even numbered ARDs were from day 2 and odd numbered ARDs were from day 4) that is higher than parental strain, confirming that these are inherited genetic effects. ARD1, 3, 5, 6, 10, and 12 were chosen for further characterization on acid resistance (Table 1). They were fully resistant to pH 3 (Fig 1F) and all displayed equal, if not higher, level of acid resistance than the D4 adapted cultures (Fig 1D and 1G). These data show that the selected ARDs (1, 3, 5, 6, 10, and 12) represent the most acid resistant survivors from the adapted cultures. Taken together, MQ140025 variants with increased acid resistance naturally emerge and they are rapidly enriched during an *in vitro* evolution experiment at low pH.

## ARD isolates display increased SigB activity

During routine laboratory culturing on BHI agar, ARD3 and ARD5 formed smaller colonies (Fig 2A) that resembles those characterized as "small colony variants" in *Staphylococcus aureus*, which are often associated with increased SigB activity [26,27]. In *L. monocytogenes*, a Δ*rsbX* strain which has a hyperactive SigB phenotype forms smaller colonies and displays reduced motility [12]. Therefore, ARD1, 3, 5, 6, 10, and 12 were assayed for motility at 30°C (Fig 2B). While the parental strain MQ140025 displayed hyper-motility (HM); ARD1, ARD3, and ARD5 were characterized by smaller colony size and low motility (LM); ARD6, ARD10, and ARD12 were characterized by normal colony size and intermediate motility (IM) (Fig 2A and 2B). Based on these observations (decreased motility and increased acid resistance relative to MQ140025), we hypothesized that the ARDs had acquired mutations to restore SigB activity to different levels. To test this, the expression of the glutamate decarboxylase system and arginine deiminase system was examined by measuring the transcript levels of *gadD3*, *gadT2*, and *aguA1* (Fig 2C–2E). Among these three genes only the transcription of SigB-dependent gene *gadD3* in the ARDs was upregulated to similar levels as reference strain F2365. No significant differences in *aguA1* transcript levels were detected between the ARDs and parental strain, suggesting no changes in ArgR activity had occurred. The GadR-dependent *gadT2* expression in MQ140025 and the

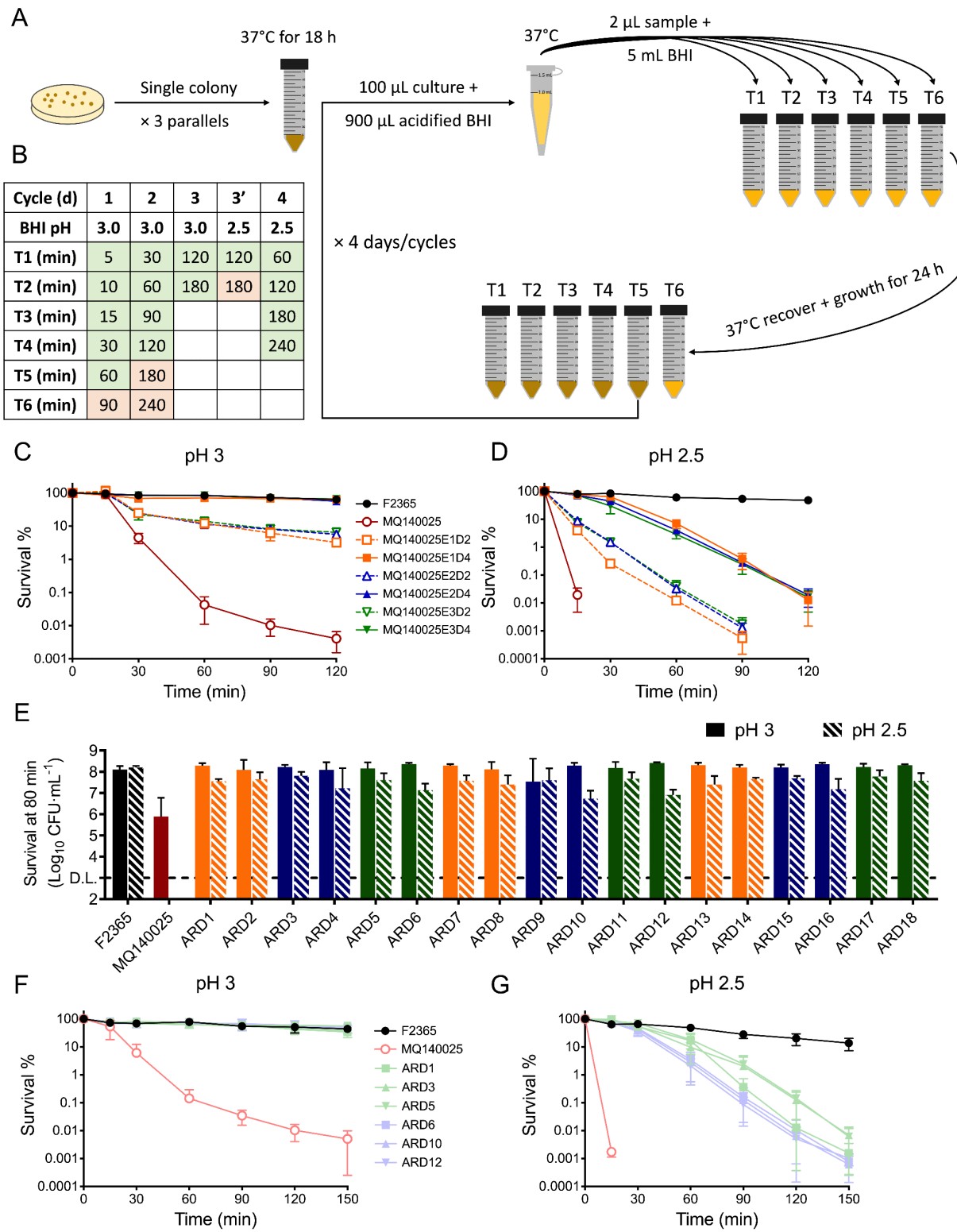

**Fig 1. Strain MQ140025 rapidly develop acid resistance during *in vitro* adaption.** Schematic presentation of *in vitro* adaption experimental setup (A), three parallel cultures of strain MQ140025 were repeatedly exposed to lethal acidic challenge and recovery cycles. The stress condition and sampling time points for *in vitro* adaption experiment are shown (B), sage and rose denotes recoverable and non-recoverable samples, respectively. The

abilities of surviving under pH 3 (C) or pH 2.5 (D) are determined for 2-day (D2) and 4-day (D4) adapted cultures from three independent experiments (E1, E2, and E3). ARDs recovered from adapted cultures were examined for their ability to withstand 80 min pH 3 or pH 2.5 challenge (E). The six ARDs selected for further characterization were determined for survival under pH 3 (F) and pH 2.5 (G). Three independent experiments were performed for all survival assay, each with technical duplicates.

**Table 1. Strains used in this study.**

| Name | SigB operon genotype | Source | Name | Source |
|---|---|---|---|---|
| *L. monocytogenes* MQ140025 strains | | | | |
| POI | *rsbX* N77K; *rsbU* Q317* | Wu et al., 2022 | ARD2 | This study (E1D2R1) |
| INT | *rsbX* N77K | This study | ARD4 | This study (E2D2R1) |
| ANC | wild type | This study | ARD7 | This study (E1D4R2) |
| ARD1 | *rsbV* G111C/[1]*rsbW* M1M | This study (E1D4R1[2]) | ARD8 | This study (E1D2R2) |
| ARD3 | *rsbW* Y37* | This study (E2D4R1) | ARD9 | This study (E2D4R2) |
| ARD5 | *rsbW* R23* | This study (E3D4R1) | ARD11 | This study (E3D4R2) |
| ARD6 | *rsbV* G111S/*rsbW* M1M | This study (E3D2R1) | ARD13 | This study (E1D4R3) |
| ARD10 | *rsbV* G111S/*rsbW* M1M | This study (E2D2R2) | ARD14 | This study (E1D2R3) |
| ARD12 | *rsbV* G111S/*rsbW* M1M | This study (E3D2R2) | ARD15 | This study (E2D4R3) |
| | | | ARD16 | This study (E2D2R3) |
| | | | ARD17 | This study (E3D4R3) |
| | | | ARD18 | This study (E3D2R3) |
| Other strains | | | | |
| *L. monocytogenes* strain F2365 | | | | |
| *E. coli* strain TOP10 | | | | |

[1]: "/" denotes effects produced by single mutation.

[2]: E1D4R1 indicate this strains was isolated as a survivor from 4 days (D4) evolved culture 1 (E1) during the 1st biological replicate (R1) of acid survival experiment (Fig 1D).

ARDs was much lower than that of F2365, similar to the transcript level observed from Δ*gadR* strains [28], suggesting no involvement of GadR in the ARD phenotypes. To corroborate this finding, the transcript levels of SigB-dependent genes *lmo2230* and *inlA* were examined. Significantly higher transcript levels of *lmo2230* and *inlA* were found in most ARDs (Fig 2F and 2G). Interestingly, *gadD3*, *lmo2230*, and *inlA* were all induced to a higher level in LM group (ARD1, 3 and 5) than IM group (ARD6, 10 and 12; Fig 2C, 2F, and 2G). This is in line with a slighter higher acid survival associated with LM group (Fig 1G). These observations indicate a positive correlation between SigB activity and acid survival and suggest that the ARDs have regained SigB activity.

**ARD isolates carry mutations in the anti-sigma factor gene *rsbW***

Phylogenetic analysis placed strain MQ140025 in the most ancient genetic clade of ST1, within a branch that underwent significant population expansion (S1A Fig). To accurately determine the genetic changes in the ARDs, complete genome sequence of strain MQ140025 was determined by long-read Nanopore sequencing. Long read genome assembly yielded three circular sequences, representing the chromosome (2902539 bp) and two plasmids (50099 bp and 11340 bp), and they were polished by Illumina sequencing (using the same DNA sample used for Nanopore sequencing). When the chromosomal sequence of strain MQ140025 was compared to CC1 reference strain F2365 (Fig 3A), no major genome rearrangement was detected except that strain MQ140025 exhibits a 3.5 kb deletion at a highly variable region encoding a type VII secretion system (T7SS) [29,30]. Close comparison between MQ140025 and F2365 revealed that: 1) MQ140025

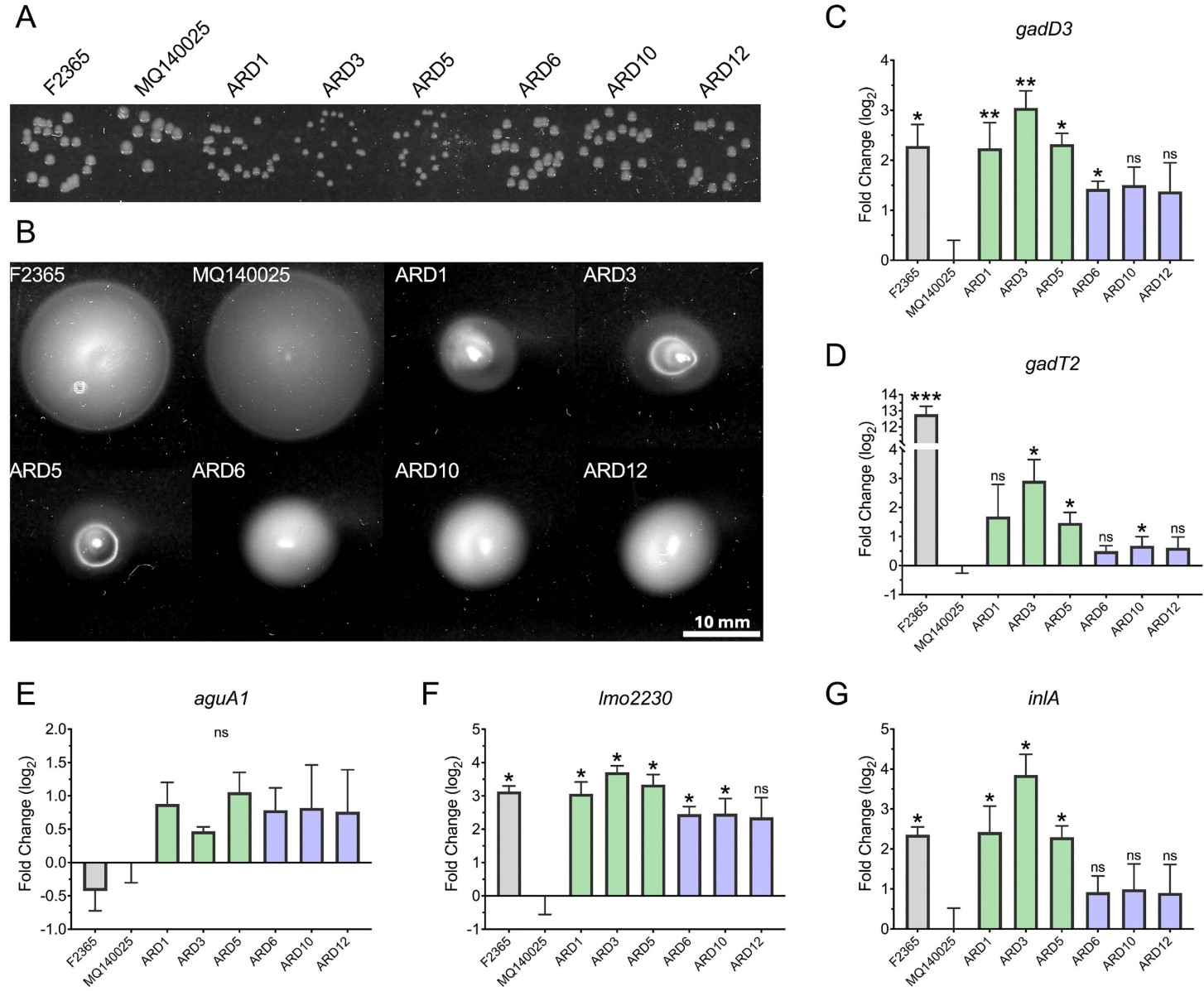

**Fig 2. ARDs display altered colony morphology, motility, and increased SigB activity.** Colony morphology after 24 h growth at 37°C (A) and motility after 48 h incubation at 30°C (B) were shown for strains F2365, MQ140025 and ARD. Transcripts levels of *gadD3* (C), *gadT2* (D), *aguA1* (E), *lmo2230* (F), and *inlA* (G) at stationary phase were measured for strains F2365, MQ140025 and ARDs and expressed relative to MQ140025 using 16S as reference gene. Three independent experiments were carried out and each with technical duplicates. Statistical significance was calculated between each strain to MQ140025 using paired two-tailed *t*-test (ns, not significant; *, $P < 0.05$; **, $P < 0.01$; and ***, $P < 0.001$).

along with other CC1 strains share premature stop codons (PMSC) with F2365 in *lmo0140*, *lmo0671*, and *lmo2084*; 2) MQ140025 doesn't bear other PMSC identical to F2365 [31]; 3) MQ140025 acquired several additional PMSCs (Fig 3A). When comparing the complete chromosome sequence of MQ140025 with the earlier acquired Illumina genome sequence (NCBI accession: GCF002027925), we observed 2 polymorphisms: *rsbU* *317Q and *lmo1979* ΔE128. This spontaneous reversion of the *rsbU* lesion to wild-type was confirmed in the MQ140025 strain used for Nanopore sequencing (by Sanger

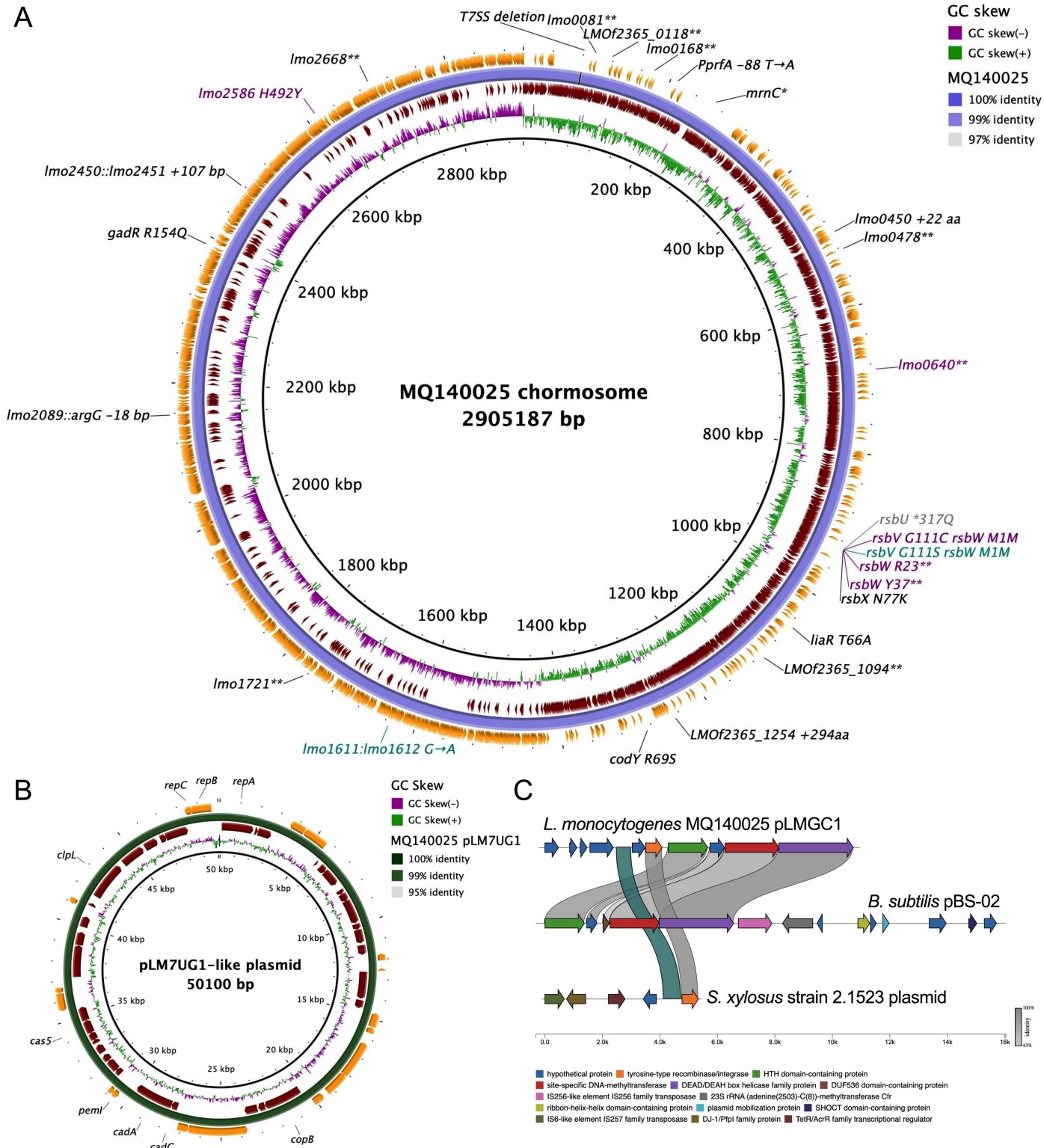

**Fig 3. Genomic characterization of MQ140025 and ARD.** Chromosomal sequence of MQ140025 was determined and mapped to reference genome sequence F2365 (A). Inner to outer rings represents: F2365 genome, genome GC skew, clockwise transcribing genes, MQ140025 genome, counter-clockwise transcribing genes. Outstanding gene disruption, substitution, insertion, and deletions (>3 bp) compared to F2365 are labelled in

black. The *rsbU* mutation that occurred during long read sequencing comparing to original short read sequencing is labelled in grey. Purple labels denote mutations detected in ARD1 (*rsbV* G111C *rsbW* M1M; *lmo2586* H492Y), ARD3 (*rsbW* R23*; *lmo0640***), and ARD5 (*rsbW* Y37**). Cyan labels denote mutation shared by ARD6, ARD10, and ARD12 (*rsbV* G111S *rsbW* M1M) and one additional mutation only in strain ARD12 at intergenic region (*lmo1611*::*lmo1612*). A pLM7UG1-like plasmid in strain MQ140025 is mapped to pLM7G1 from *L. monocytogenes* serotype 7 strain SLCC2482 (B). Genomic architecture of a novel plasmid pLMGC1 is shown along with *Bacillus subtilis* plasmid pBS-02, and an unnamed *Streptococcus xylosus* plasmid (C). Sequence homologies within open reading frames are indicated in grey, and sequence homology detected in intergenic region is indicated in blue.

sequencing). This highlights the plasticity of loci involved in the GSR during routine laboratory passage, and provided an opportunity to investigate the effect of the *rsbU* Q317* lesion on SigB activity. To discriminate these two MQ140025 strains, they are renamed as strain "POI" (Point Of Isolation, carrying *rsbU* Q317* and *rsbX* N77K) and strain "INT" (INTer-mediate, carrying only *rsbX* N77K Table 1). Of the two plasmids found in MQ140025, the larger one is virtually identical to a prevalent plasmid pLM7UG1 in *L. monocytogenes* (Fig 3B) [32]. The smaller plasmid does not resemble any plasmid previously reported from *Listeria*, although Blastn found regions that share similarity to a known *Bacillus subtilis* plasmid (pBS-02) and an unnamed *Streptococcus xylosus* plasmid (accession: CP066723). We assign the name pLMGC1 to this newly identified plasmid (Fig 3C).

To elucidate the genetic changes in the ARDs, Illumina sequencing reads from ARD1, 3, 5, 6, 10, and 12 were mapped to the MQ140025 genome. Surprisingly, mutations were found in *rsbW* from all ARDs. ARD3 and ARD5 each carries a distinct non-sense mutation in the *rsbW* coding sequence (Fig 3A), resulting in truncated variants of RsbW. Interestingly, all other ARDs carried SNPs affecting the *rsbW* start codon (SC). Since the coding sequences of *rsbW* and *rsbV* overlap (17 bp), these mutations produce dual effects: they substitute RsbV glycine at position 111 with cysteine (ARD1) or serine (ARD 6, 10, 12) and they also change the *rsbW* start codon from a canonical ATG start codon (SC) to a non-canonical ATT (ARD1) or ATA (ARD 6, 10, 12) SC (Fig 3A). We reasoned that this could perturbate either the function of RsbV or, more likely the translation rate (and thus level) of RsbW. Although additional mutations were detected in ARD1, ARD3, and ARD12, they do not phenotypically distinguish these strains from other ARDs baring similar *rsbW* mutations. Therefore, we conclude it was the mutation within *rsbW* that led to the increased SigB activity and elevated acid resistance.

## Sequential acquisition of *sigB* operon mutations impact fitness

To elucidate the individual roles of *sigB* the operon lesions (*rsbU* Q317* and *rsbX* N77K) in the phenotype of the clinical isolate, we took advantage of the INT strain and further genetically repaired its *rsbX* N77K lesion, generating an ances-tor strain of MQ140025 (ANC) with wild type *sigB* operon genotype. The ANC strain produced a normal colony size and displayed equal motility to the reference strain F2365, suggesting that the lesions in the *sigB* operon perturbed SigB activity with consequences for growth and motility. The N77K substitution in the negative SigB regulator RsbX caused a reduction of both colony size and motility in strain INT, likely resulting from a higher SigB activity (Fig 4A). While the pre-mature translation termination (Q317*) of the positive SigB regulator RsbU produced the opposite effect (Fig 4A). These observations suggest that *rsbX* N77K and *rsbU* Q317* are both loss-of-function mutations. Growth experiments confirmed a fitness defect in INT strain (caused by *rsbX* N77K) and a fitness advantage in POI strain (caused by *rsbU* Q317*) as measured by growth rate in BHI media under 30°C (Fig 4B and 4C). However, this difference was not significant under 37°C (Figs 4C and S2A). To consolidate these findings, the survival of these strains under acid stress was measured using stationary phase cultures. The strain INT survived the best, followed by strain ANC, while strain POI displayed weakest acid resistance (Fig 4D). In line with this observation, the transcript levels of SigB-dependent genes *inlA*, *gadD3*, and *lmo2230* were the highest in strain INT, and the lowest in strain POI (Figs 4E, S2B, and S2C).

To explore the biological significance of the *rsbW* mutations, we compared the fitness and SigB activities of the ARDs with POI, INT, and ANC strains. ARD1 (*rsbW*ᴬᵀᴳ-*rsbW*ᴬᵀᵀ), ARD3 (*rsbW**), and ARD6 (*rsbW*ᴬᵀᴳ-*rsbW*ᴬᵀᴬ) all displayed reduced motility while ARD1 (*rsbW*ᴬᵀᴳ-*rsbW*ᴬᵀᵀ) and ARD3 (*rsbW**) also formed small colonies (Fig 4A). Interestingly, they

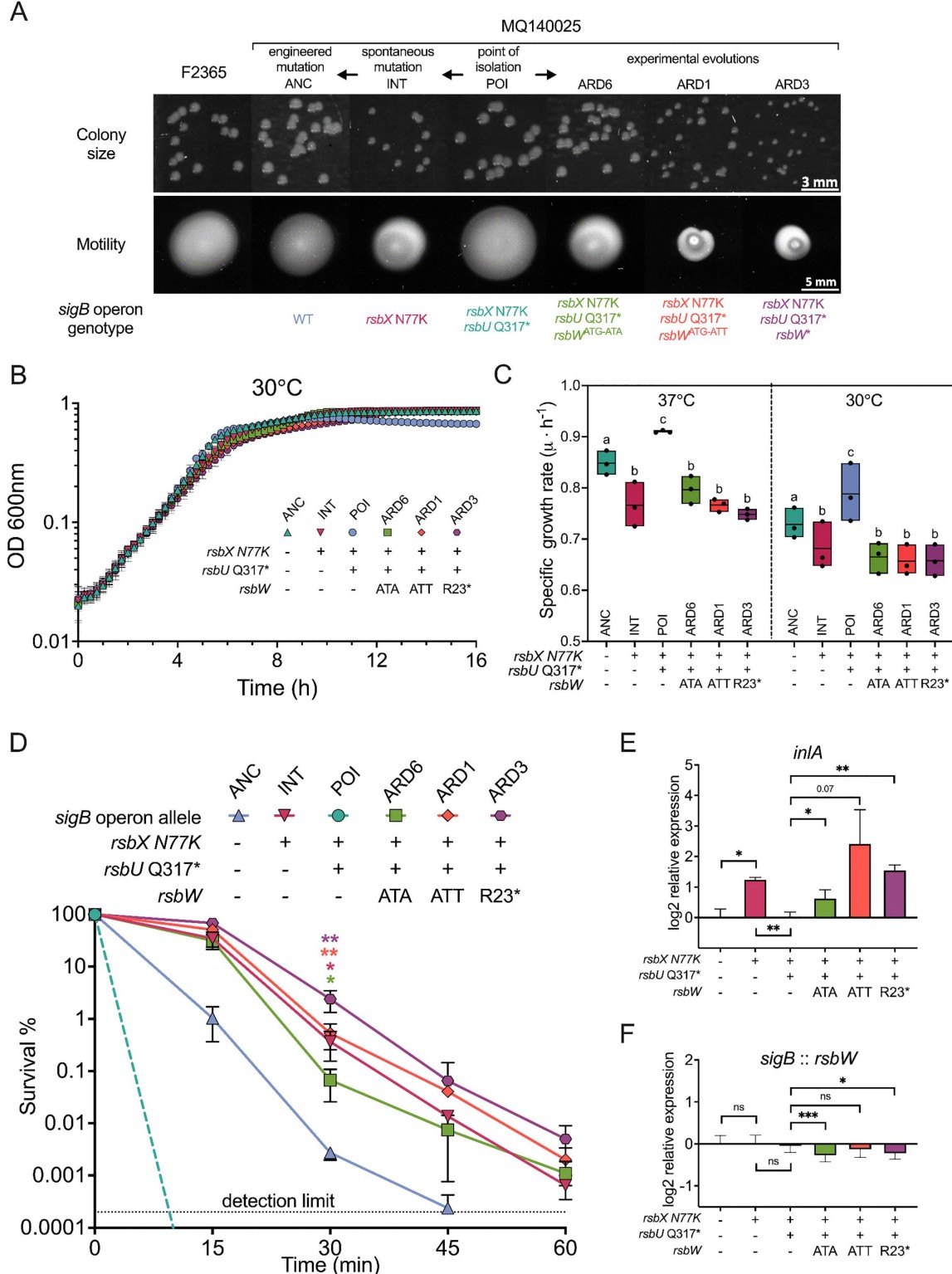

**Fig 4. Sequential acquisition of *rsbUWX* mutations influence fitness by altering SigB activity.** Colony morphology at 37°C and motility at 30°C were assayed for strains of MQ140025 after 24 h incubation (A). Growth curves at 30°C (B) and maximum growth rates measure using growth curve from 37°C and 30°C (C) are shown. Statistical differences were calculated using one-way ANOVA (*P* < 0.05). Percentage survivals of stationary phase

cultures during pH 2.5 treatment are shown for various strains of MQ140025 (D). For growth and survival assays, three independent experiments were carried out and each with technical triplicates. Transcripts levels of *inlA* at stationary phase were measured and expressed relative to MQ140025 strain with wild type *sigB* operon using 16S as reference gene (E). Transcripts levels of *sigB* at stationary phase were measured and expressed relative to MQ140025 strain with wild type *sigB* operon using *rsbW* as reference gene (E). For transcriptional analysis, three independent experiments were carried out and each with technical duplicates. All statistical significance was calculated between each strain to MQ140025 using paired two-tailed *t*-test (ns, not significant; *, $P<0.05$; **, $P<0.01$; and ***, $P<0.001$).

all displayed reduced growth rates at 30°C comparing to the ANC strain. These observations are further supported by the increased acid resistance and elevated transcript levels of SigB-dependent genes in the ARDs (Figs 4D, 4E, S2B, S2C, and S2D). As decendents of strain POI (with reduced SigB activity), the fitness and transcriptional profiles of ARDs are reminiscent of that observed in the INT strain (with increased SigB activity), demonstrating the dominant effect on SigB activity of the *rsbW* mutations. Overall, these results suggested that *sigB* operon lesions likely occurred sequentially in strain MQ140025 upon occurrence of specific environmental selective pressures, and that the *rsbW* mutations circumvent the previous lesions within *sigB* operon to directly activate SigB.

## Non-canonical SCs reduce *rsbW* translation efficiency

Next we sought to understand the mechanism of SigB activation in ARD1, ARD6, ARD10, and ARD12. We reasoned that any effect produced by these SNPs on *rsbW* is likely to be prominent since RsbW acts downstream of RsbV by directly interacting with SigB. It is noteworthy that, despite the upregulation of the *rsbW-sigB* transcript in the ARDs, the ratio between *rsbW* and *sigB* transcripts was maintained (Fig 4F). Given the antagonistic effect of RsbW on SigB activity, it is likely that the *rsbW* transcript was translated less efficiently than the *sigB* transcript, leading to a perturbed RsbW::SigB stoichiometry. The presence of non-canonical SCs, albeit decoding as methionine, had the potential to reduce the RsbW translation efficiency. In strain EGD-e a canonical ATG was confirmed as the primary SC of RsbW [33], of note is the existence of a secondary SC of *rsbW* 6 bp downstream (Fig 5A). Since the transcription of *rsbW-sigB* is in part autoregulated by SigB, we used a SigB-independent promoter to test the influence of the non-canonical SCs on the RsbW translation efficiency. For this, we designed three translational reporter constructs using the IPTG-inducible integrative expression vector pIMK3 [34]. In order to preserve the native ribosome binding site (RBS) of *rsbW*, the 3'-end of *rsbV* sequence was fused to codon-optimized eGFP [35] to produce a 16 aa peptide tail of RsbV and a RsbW-eGFP fusion protein (Fig 5A). Three reporters were constructed using ATG, ATA, or ATT as the primary SC for *rsbW-eGFP* fusion proteins, and they were introduced to MQ140025 strain POI with pIMK3 as control. The induction of a fluorescence signal by IPTG was clearly decreased when using ATA as the SC compared to ATG, while there was no detectable signal when using ATT as SC (Fig 5B). These data suggested that the relative translation rates from these three start codons were ATG>ATA>ATT. We then performed Western-blots using anti-eGFP antibodies on crude protein extracts from the same culture conditions. While eGFP expression was detected from both the ATA and ATT constructs it was at a reduced level compared to the canonical ATG SC, with the ATT construct again showing the lowest translation rate (Fig 5C). These results suggest that the primary SC of *rsbW* accounts for the functional expression under laboratory conditions, and the usage of an efficient SC is critical for RsbW expression. This supports a model in which ARD1, ARD6, ARD10 and ARD12 achieve elevated SigB activity through an altered translation rate of RsbW, which in turn is predicted to perturb the RsbW:SigB stoichiometry (Fig 5D).

## Canonical SC are highly preferred for genes within *sigB* operon

The strong influence of *rsbW* start codon selection on SigB activity prompted us to investigate whether SC selection could play a general role in fine tuning SigB activity across the species. To investigate this, we compared the SC usage in the 8 genes comprising the *sigB* operon from 60,690 *L. monocytogenes* whole genome sequences available from the NCBI

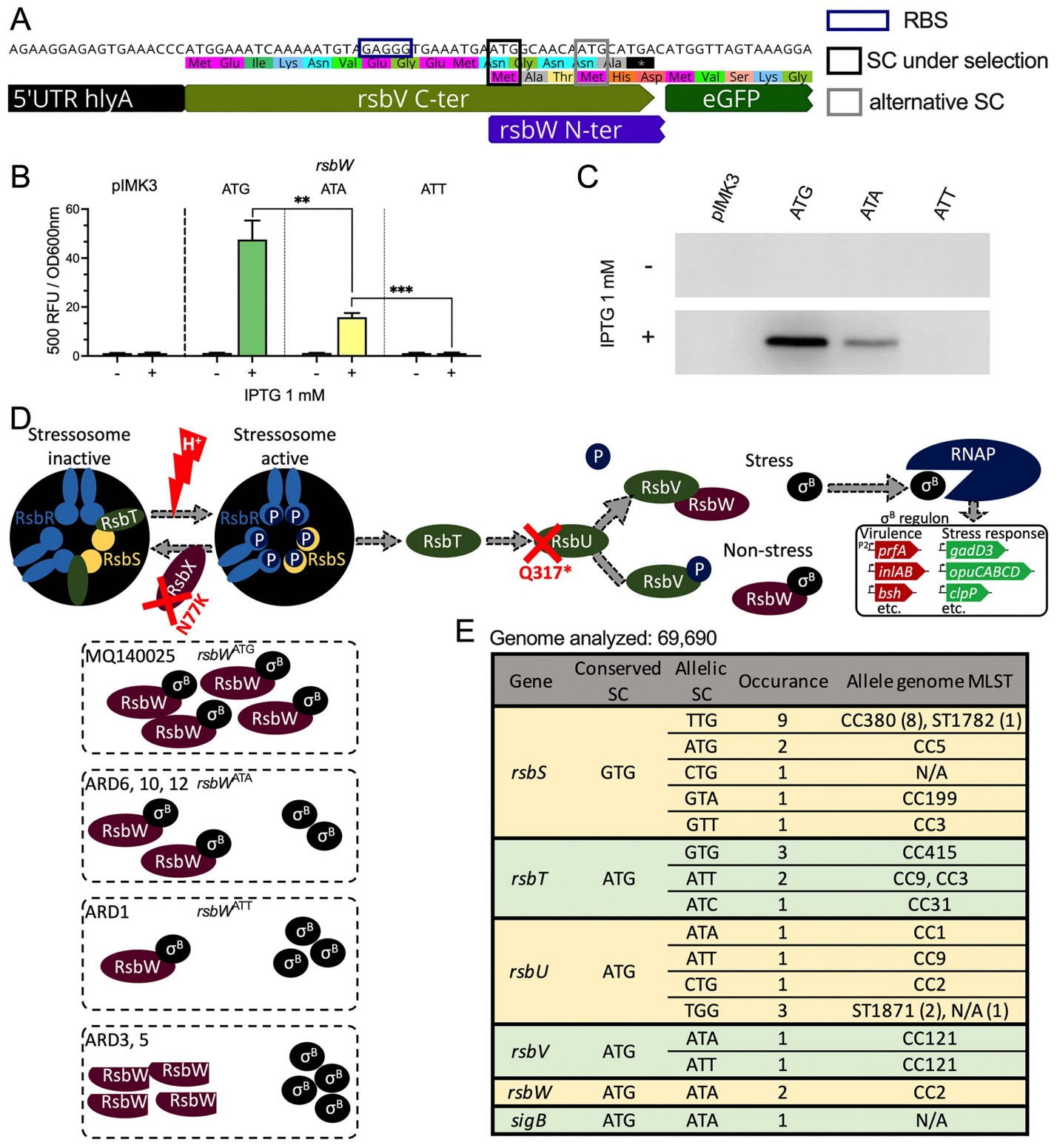

**Fig 5. ARDs rescue SigB activity by reducing *rsbW* translation or abolishing RsbW function.** Schematic presentation of translational reporter design for pIMK3 *rsbW*-ATG is shown (A), with 5'UTR *hly* partially omitted. Primary and secondary SCs of *rsbW* are outlined by black and grey rectangular, respectively. Normalized fluorescent signals are shown for *rsbW* translational reporter strains from stationary growth phase (18 h) with or without

1 mM IPTG (B). Western-blot quantification of GFP expression in reporter strains in the absence or presence of 1 mM IPTG (C). Schematic presentation of SigB activation pathway in strain MQ140025, and proposed modes of actions by which SigB is (in)activated in MQ140025 and ARD are shown (D). Occurrences of SC alterations in *sigB* operon by analysing 60 K *L. monocytogenes* genomes are shown (E). Statistical significance using paired two-tailed *t*-test (ns, not significant; *, $P < 0.05$; **, $P < 0.01$; and ***, $P < 0.001$).

database. All *sigB* operon genes use ATG as a SC except RsbS (GTG), and this is generally conserved at the species level (Fig 5E). Translation initiation from GTG SCs was recently shown to be weaker than ATG in bacteria [36]. It seems plausible that GTG SC usage for *rsbS* might be a means of achieving the correct stressosome subunit stoichiometry, which is reported to be 2:1 for RsbR:RsbS [8,37,38]. Overall the alteration of SCs in the *sigB* operon occurred at a very low frequency (n = 31, ~0.05%, Fig 5E). In two closely related strains (both CC2), SC mutations in *rsbW* (ATG→ATA) were observed (Fig 5E), suggesting an environmental selective pressure that favours high SigB activity. It is noteworthy that 8 out of 9 GTG→TTG SC substitutions in *rsbS* were from a rare genetic clade, CC380 (Fig 5E). Taken together, canonical SC are generally highly preferred in the *sigB* operon suggesting that efficient translation initiation is important for the integrity of the GSR.

## SC are under selection for stress response and virulence genes

To systematically analyse the role of SC usage in the evolution and genetic adaption of *L. monocytogenes*, we extended the analysis from the *sigB* operon to the entire core-genome. We identified 2265 conserved genes among 60,690 *L. monocytogenes* genome analysed. The majority (98.2%; n = 2226) of these genes use an unchanging canonical start codon (ATG: 81.5%; TTG: 10.2%, GTG: 8.0%, S1 Table). Interestingly, fixed usage of non-canonical start codons was found for six conserved genes (S1 Table). Among these six genes, the translation initiation sites of *infC* (ATT) and *rnhC* (ATC) have been experimentally confirmed in the reference strain EGD-e [33]. SC usage was found to be flexible for 39 conserved genes (1.7%). Seventeen of these were experimentally confirmed previously, while the remainder have not yet been experimentally tested (Fig 6A). Interestingly, one third of these 39 genes (n = 13) are known to be involved in virulence and/or stress response (Fig 6A). *esaB* encodes a cytoplasmic protein of a T7SS, which negatively influences virulence [30,39]. *plcB* encodes phosphatidylcholine-specific phospholipase C that is required for cell-to-cell spread during infection [40]. *lmo1402* is co-transcribed with *lmo1400* and *ymdB* (*lmo1401*), and disruption of *lmo1401* by transposon results in reduced haemolytic activity [41]. *comEC* encodes a late competence protein that is involved in phagosome escape [42]. Lmo1638 is a putative carboxypeptidase and it was shown to play a role during later stage infection [43]. *lmo2445* encodes an internalin-like protein within an α-glucan metabolism operon that is required for full virulence [44]. Lmo0669 encodes a stress inducible SigB-dependent oxidoreductase. *lmo0501* (*mltR*) encodes transcriptional regulator for mannitol metabolism that promotes cold, osmotic, and acid stress response [45]. Lmo0093 (KtrD) is a subunit of low affinity potassium transporter KtrCD which plays a role in osmolyte homeostasis [46]. *lftR* encodes a transcriptional repressor that influences virulence and antibiotic resistance [47,48]. MntC is a subunit of ABC-type manganese transporter which contributes to pathogenicity and stress response [49,50]. Lmo0946 (Sif) was recently shown to contribute to β-lactam antibiotic susceptibility, GSR and virulence [51]. The majority of these genes use different SCs between lineage I and lineage II, and some of them use different SCs between globally prevalent clonal complexes (S2 Fig). These data suggest an evolutionary preference for the usage of specific SC for these genes possibly associated with niche-specific selective pressures.

While the majority of SC flexibility in these 39 conserved genes are within canonical SCs, the level to which altered canonical SCs influences gene expression is unknown in *L. monocytogenes*. To test this, we designed eGFP-based translational reporters to compare the translation efficiency when using ATG or GTG as SC within the genomic context of *plcB*

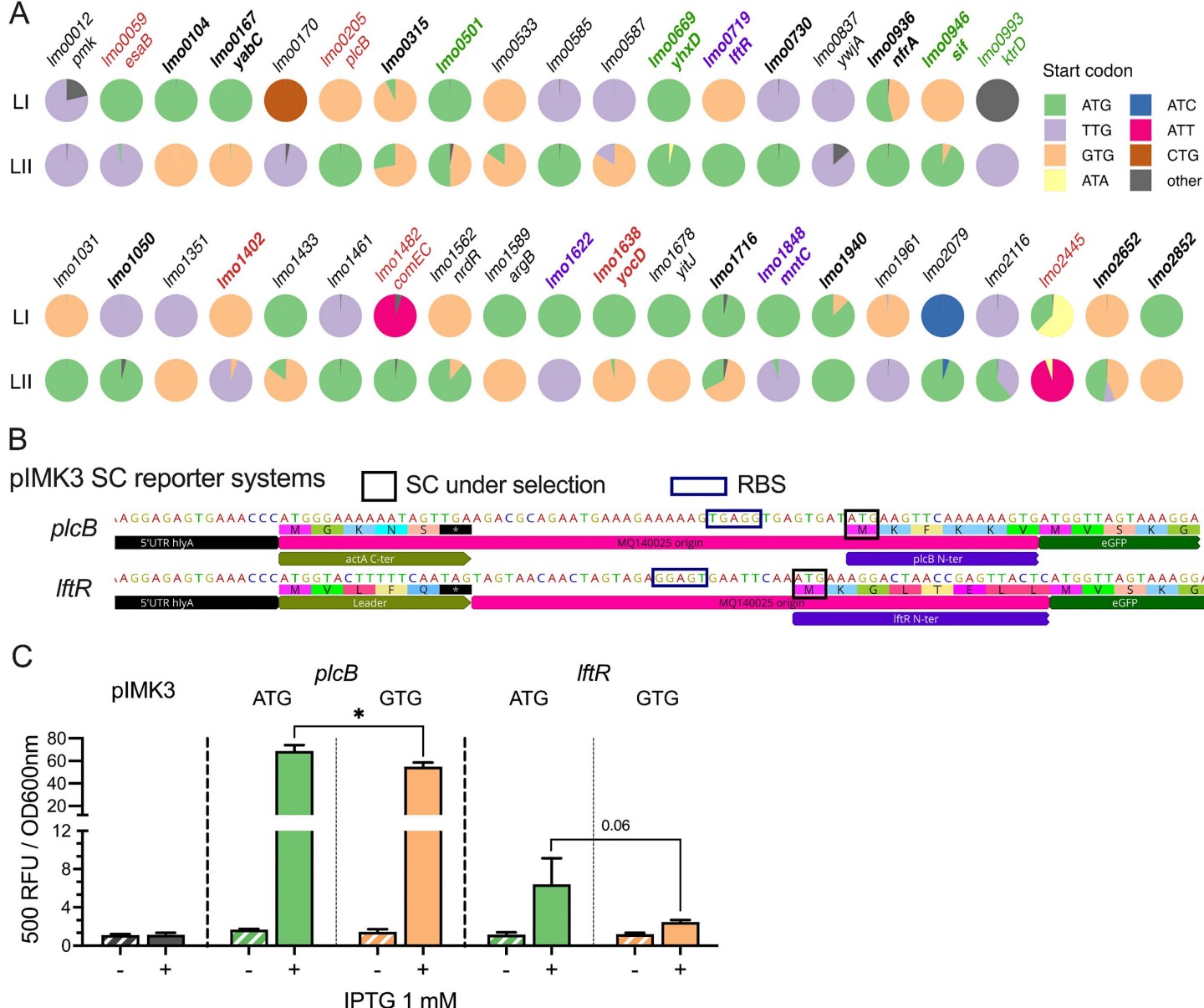

**Fig 6. SC selection as a means of modulation for stress and virulence genes expression.** The SC usage in 39 conserved genes with flexible SC is shown for lineage I (LI) and II (LII) *L. monocytogenes* strains (A). The names of genes that are involved in stress response, virulence, or both are presented in green, red, and purple, respectively. The genes with experimentally confirmed SC are in bold [33]. Schematic presentations of translational reporter design for SC activity in *plcB*, *lftR*, and *mntC* are shown (B). Normalized fluorescent signals are shown for each reporter strain from stationary growth phase (18 h) with or without 1 mM IPTG (C). Statistical significance using paired two-tailed *t*-test (ns, not significant; *, *P* < 0.05; **, *P* < 0.01; and ***, *P* < 0.001).

and *lftR* (Fig 6B). In both cases, a strong fluorescence signal was detected when using ATG as start codon, while a small reduction in the translation rate was detected when GTG was present (Fig 6C). These results suggest that for some genes SC flexibility might provide a genetic means of adjusting gene expression, potentially as a means of enhancing niche specific fitness.

## Discussion

In this study, we characterized the genome of clinical *L. monocytogenes* strain MQ140025 and report its rapid genetic adaptation to extreme acid stress. This adaption occurred through increasing SigB activity by genetically disabling its antagonist, RsbW. Interestingly, in several adapted mutants this was achieved by using non-canonical SCs for *rsbW* translation initiation. This prompted us to investigate the overall SC usage in *L. monocytogenes* and the potential effect of SC usage on gene expression. These findings highlight the genetic plasticity of *L. monocytogenes* when under extreme selective pressure from environmental stress and demonstrate that alternative SC usage can be an effective means of regulatory control during niche adaptation.

Stress responses occur during bacterial adaptation to ever-shifting environments. *L. monocytogenes* as a food-borne pathogen colonizes a wide range of niches, and understanding its stress adaptation is important for its successful control. Here, we applied extreme selective pressure by repetitively recovering the most acid resistant populations from parallel bacterial cultures. This was designed to mimic the bottleneck events in gastric fluid passage during infection, where the acid resistant bacteria are more likely to survive and access the intestine. The most acid resistant cells from the adapted cultures were isolated during this acid survival experiment. These strong selective pressures explain the enrichment and occurrence of rare SigB hyper-active mutations [22], which come with fitness burdens during growth (Fig 2A). It is likely that mutants with a range of SigB activities are present in each acid adapted culture, but only those with the highest resistance were selected for by the conditions. Interestingly, the strain POI and the ARDs strains displayed little expression of glutamate decarboxylase system encoded by *gadT2D2* (Fig 2D) [52], which is controlled by the recently characterised RofA-like transcriptional regulator GadR [28]. A careful examination of the GadR sequence in MQ140025 revealed a substitution (R154Q) in its DNA binding domain. GadR-mediated *gadT2D2* expression is critical for survival at pH 2.4 and this defect in *gadT2D2* expression likely explains the reduced acid resistance in the ARDs when compared to the closely related strain F2365 (which encodes an otherwise identical GadR). It is noteworthy that when we used strain EGD-e (which carries a premature stop codon (L374*) disrupting the *gadR* gene [28] in a similar acid adaption experiment but at a lower pH, it restored GadR activity after only two inactivation-recovery cycles by restoring *gadR* full length open reading frame (*374K). Thus, increased SigB activity provides strain MQ140025 significant selective advantage for surviving pH 3 and pH 2.5 but it is likely that more severe acid stress would be necessary to select for increased *gadT2D2* expression. These data suggest that the severity of the stress applied substantially shifts the types of genetic adaptations selected during *in vitro* evolution experiment.

Phylogenetic analysis of MQ140025 allowed examination of the evolutionary trajectory of MQ140025. We compared the core genome mutations associated with bottlenecking events and population expansion events historical to clinical strain MQ140025. Interestingly, GadR R154Q occurred much earlier on in the strains ancestral to the population expansion events (S1B Fig). The subsequent population expansion event was associated with Lmo0884 E111D and CodY R69D. Given the conservative nature of former (E111D) substitution, we reasoned that the CodY substitution (R69D) most likely confers the predicted fitness alteration. CodY is an important global regulator that controls amino acid metabolism, stress response and virulence in response to GTP and branch chain amino acids [53]. R69 resides within the isoleucine binding box of CodY and conceivably its substitution would impact isoleucine responsiveness and therefore overall fitness [54,55]. Strain MQ140025 emerged within this clade immediately following a further population expansion events, associated with four core-genome mutations including LiaR T66A and P*prfA* -88 T→A. LiaR is response regulator that mediates cell envelope stress response and the translation of PrfA is subject to extensive post-transcriptional control in 5'-UTR [56,57]. The *sigB* operon mutations (*rsbU* Q317* or *rsbX* N77K) were found specific to strain MQ140025. This suggested that these are transient mutations that confer niche-specific advantages. Using spontaneous mutants and genetically engineered mutants, we showed that they are both loss-of-function mutations. As RsbU protein functions downstream to RsbX in the SigB activation cascade (Fig 5D), it is unlikely that *rsbX* N77K mutation was selected after the *rsbU* Q317* mutation arose. Given the clinical relevance of this strain, this observation supports our previous hypothesis [21] where

it is likely that the GSR-activating mutation *rsbX* N77K was firstly selected upon host-entry, where gastric acid was likely encountered. Subsequently, the GSR-disabling mutation *rsbU* Q317* occurred due to the growth burden associated with a high GSR (Fig 4C). These observations highlight the changes within the core-genome during evolutionary selection that likely influence the stress response and virulence of the pathogen [18,28,58].

The emergence of acid resistant mutants in the MQ140025 background after only 2 days of selection highlights the capacity for rapid genetic adaptation to extreme environmental stress. The finding that 6 independent isolates from this short-term IVEE were confirmed by whole genome sequence analysis to carry newly acquired lesions in the *sigB* operon demonstrates the importance of GSR to acid resistance, and therefore host entry. Previous studies from our group and others have reported on the prevalence of loss-of-function mutations in the *sigB* operon in public databases, under mild-stress or during laboratory culture [20–22,59]. Interestingly, we have also previously observed a *rsbS* SC alteration from a *sigB*⁻ mutant following *in vitro* evolution experiment [20]. Collectively these studies highlight the plasticity of the *sigB* operon and suggest that environmental stress can be a significant driver of evolution in this pathogen. Gain of function mutations (that increase SigB activity) are not commonly reported, likely because these mutations are expected to come with a fitness cost as they can negatively impact cell growth and reproduction [12,60]. The *rsbW* mutations reported here are predicted to increase SigB activity by affecting the stoichiometry of RsbW to SigB, the principle interaction that limits SigB availability for participation in transcription. Interestingly, a recent study also observed sequential occurrences of loss-of-function mutations in *rsbU* and *rsbW* from evolved biofilm-associated mutants [61], suggesting that the genetic adaption we report here may not be rare. Thus, when a severe environmental stress (lethal acid pH) is encountered the survival advantage gained through increasing SigB activity likely outweighs any fitness cost associated with this change, at least in the short term. The finding that *rsbW* non-canonical SCs are rare (2 found in 60,690 genomes, Fig 5E) in the reported genome sequences of *L. monocytogenes* suggests that in the long-term the cost does indeed outweigh the short-term survival advantage. Interestingly, one study that investigated the genetic adaption of *L. monocytogenes* to repeated murine passage following oral inoculation found that PMSCs frequently arose in the *rsbW* gene [62], perhaps indicating that the conditions in the mouse GI tract can provide significant selective pressure for the loss of RsbW function. A recent study has confirmed that there is a strong positive correlation between hyper-virulence and SigB activity, suggesting the host environment provides significant selective pressure for altered SigB activity [18]. Future research will investigate whether and how *sigB* operon mutations differ in their distribution among clinical and environmental isolates, which may give insights into the precise environmental selective pressures that drives the occurrence of such genetic adaptions.

The appearance of non-canonical SCs in *rsbW* in the acid resistant isolates raises the possibility that a reduced translation rate is serving a regulatory function in these strains (i.e., to increase SigB activity). When the start codon usage was examined across the entire genome for all available > 60K genome sequences (via NCBI) a number of interesting points emerged. The vast majority of ORFs in the core genome use either ATG (81.5%), TTG (10.2%) or GTG (8.0%) as start codons (S1 Table), with less than 0.3% using rare non-canonical SCs (ATA, ATC, ATT, CTG) or flexible SCs. When altered SC usage occurs for a given ORF it tends to occur along phylogenetic lines, either within an entire lineage or clonal complex. For example, the *argB* gene (*lmo1589*) is initiated by ATG in lineage I but by GTG in lineage II stains, whereas *lmo0315* is initiated by GTG in most clonal complexes, except for CC8, CC204, and CC121 where ATG is preferred (S2 Fig). This fact suggests that changes in SC usage may have been selected at some point, potentially offering some niche-specific advantage. The most likely route for a selectable advantage to altered SC stems from the impact of the SC on translation initiation rate. A recent study has shown that non-canonical SCs can provide a selective advantage in *E. coli* in the murine gut environment. When the initiation codon of the *lacI* gene, which encodes a repressor for lactose utilisation, was altered to GTG the translation rate of *lacI* was reduced thereby enhancing lactose consumption [36]. A systematic study of SC efficiency in *E. coli* showed that in addition to ATG, GTG and TTG, which had the highest initiation rates (in that order), CTG, ATA, ATT and ATC were also functional as SCs, albeit with reduced rates [63]. While the mechanism of translation initiation control in *L. monocytogenes* is not well studied it appears to be distinct from other well studied

bacteria in that the RBS strength does not correlate closely with translational efficiency [64]. The SC data presented in this study suggest that non-canonical SC selection could be a means of regulation for the expression of some genes, including those playing a role in virulence and stress resistance.

The variable usage of SCs in 39 conserved genes from different genetic clades raises the question of what are the evolutionary pressures driving these selections? Although the core genome content of *L. monocytogenes* species is highly conserved (despite two major lineages), some genetic groups (e.g., CC1, CC2, CC4, CC6, CC87) are more virulent than others (CC121, CC9, CC18, CC37) [18]. While differences in the accessory genome don't fully account for hypervirulent phenotypes [65], it is now evident that polymorphisms in the core-genome contribute to differential stress response and virulence between hypervirulent and hypovirulent strains/clades [18,66]. Interestingly, the disruption of the *lmo1622* CDS was recently shown to increase SigB activity in a hypervirulent strain [18]. Here we report the divergent SC usage of *lmo1622* plausibly selected concurrently with the branching event that produced lineages I and II (Fig 6), suggesting fine-tuning of *lmo1622* expression might be associated with evolutionary niche-specific adaption. More evidence suggesting a role for alternative SC usage in virulence was found here (Fig 6) for phospholipase C (PlcB) [40], competence component (ComEC) critical for intracellular growth [42], as well as cell invasion and environmental persistence related regulator (LftR) [47,48,67]. While changing between canonical SCs might not alter expression drastically, the fact that they are maintained within genetic clades suggests that they likely contribute to an overall genomic shift that provides an incremental fitness advantage during novel niche colonization. Our future research will focus on a detailed analysis of how these genetic discrepancies contribute to the overall differences in stress response and virulence between *L. monocytogenes* lineages. Deciphering the driving forces of these adaptations will be a challenge for future research in this area but it should contribute to the overall understanding of the biology of this important human pathogen.

## Materials and methods

### General growth conditions

*L. monocytogenes* frozen stocks were prepared using overnight cultures in BHI growth medium with 7% DMSO and stored at -80C°. For stationary phase culture, *L. monocytogenes* colonies were inoculated into 5 mL BHI (LAB M LAB048) in 50 mL conical centrifugation tubes and incubated for 18 h at 37C° with agitation (160 rpm). To revive acid stress adapted *L. monocytogenes* cultures, -80C° stocks were revived in BHI at 37C° for 24 h, then 2 µL culture was used for inoculating overnight cultures (18 h at 37C°). Where specified, kanamycin (50 µg/mL) was supplemented to growth media. For the motility assay, BHI with 0.25% (w/v) agar was seeded with 2 µL of overnight culture and incubated for 24 h or 48 h at 30°C before pictures were acquired. For growth measurements, *L. monocytogenes* cultures grown to stationary phase were washed with BHI then inoculated to fresh media to achieve OD600nm of 0.05. After inoculation, these cultures were incubated in 96-well plates (200 µL per well) and incubated in a shaking microplate reader (BioTek Synergy H1) with growth measured every 15 min. The growth rates were calculated for each one-hour window and the highest value was taken to represent the maximum growth rate for each technical repeat. The average growth rates was calculated as means of three independent experiments, each was calculated as mean of six technical repeats.

### *in vitro* acid adaption and acid survival experiments

Protocols for acid challenge experiments were adapted from a previous study [28]. Briefly, 100 µL overnight culture was mixed with 900 µL acidified BHI (pH 2.5 or pH 3) and incubated at 37°C statically. For *in vitro* adaption experiments, at each indicated time point 2 µL of sample was removed and diluted into 5 mL BHI unacidified and incubated at 37°C with agitation (160 rpm) to recover for 24 h. For acid survival experiments, at each indicated time point 20 µL sample was taken and serially diluted in phosphate saline buffer. 10 µL from each dilution was spotted on BHI agar and incubated for 48 h before colony counts were recorded. The percentage survival at each time point was calculated as the colony counts divided by initial colony counts. All experiments were carried out three independent times, each with technical triplicates.

## Whole genome sequence analysis

To characterize the complete genome sequence, one colony of strain MQ140025 revived in tryptic soy agar was used to prepare overnight culture in tryptic soy broth (18 h at 37C°). Genomic DNA was extracted using Qiagen DNeasy Ultra-Clean Microbial Kit according to manufacturer's instruction. A library was prepared using rapid barcoding kit V14 (SQK-RBK114.24, Oxford Nanopore) following instructions. The constructed DNA library was loaded onto a PromethION 2 Integrated (P2i) sequencer (Oxford Nanopore) for sequencing and real-time base calling with MinKNOW (high-accuracy option). Yielding read files were first combined and then binned for genome assembly using Autocycler [68], by compressing the result from individual assemblers (canu, flye, metamdbg, and unicycler). The resulting genome sequence was annotated using Prokka [69]. Chromosome and plasmid comparison and visualization were carried out in BRIG [70] and geneviewer (van der Velden N 2025). The complete genome sequence of strain MQ140025 was deposited at NCBI database (Project accession: PRJNA1305691).

To characterize the gain or loss-of-function mutations in strain MQ140025, chromosomal sequences from MQ140025 and F2365 were carefully compared. Firstly, the amino acid sequences from all genes in F2365 were extracted and used as query sequences to blast (tblastn) in MQ140025 chromosome. These tblastn results (hits) were examined when query coverage is less than 100% to check for premature stop codons. Then the two chromosomes were compared using Mauve to check for gaps in the alignment. These gaps present in coding sequences were manually examined for reading frame disruption or for significant insertion or deletion. The gaps greater than 10 bp within intergenic regions were also marked for further notice. Lastly, the loci where authentic premature stop codons were previously reported in strain F2365 were examined in strain MQ140025, along with other CC1 strains and reference strain EGD-e. In addition, the long read-assembled MQ140025 genome sequence was also compared with previously published short reads-assembled genome to check for mutations that arose during laboratory passages.

To analyse derivatives of strain MQ140025, genomic DNA was extracted as previously described [21]. Briefly, 5 mL of an overnight culture was resuspended in 180 μL enzymatic lysis buffer (20 mM Tris-HCl, 2 mM EDTA, 1.2% Triton X-100, and lysozyme 20 mg mL$^{-1}$) and incubated at 37°C for 30 min. Protease K and Buffer AL (Qiagen DNeasy Blood and Tissue kit) were then added to the samples and heated up to 56°C for 30 min. The remaining procedures were carried out following the manufacturer's instructions (Qiagen DNeasy Blood and Tissue kit). Purified DNA samples were sent to Novogene and the resulting pair-end sequencing reads were mapped to the MQ140025 genome (Breseq) to check for mutations [71].

## Phylogenetic analysis

To analyse the phylogeny of ST1 strains, the multi-locus sequence type was determined for the aforementioned 66,690 genomes [72]. In total 5546 genomes were classified as ST1. The phylogeny of these ST1 genomes were inferred using Parsnp2 based on the core-genome SNP profile [73]. The SNPs associated with branching points of interests were examined using Parsnp2 alignment output.

## Molecular methods

The codon-optimized eGFP sequence [35] was amplified by PCR using primers that were designed to include the regulatory regions under investigation (Table 2). These fragments along with pIMK3 plasmid were digested (NcoI and BamHI, FastDigest, Thermofisher) and ligated then transformed to *E. coli* TOP10 to propagate. Then purified recombinant plasmids were electroporated into electrocompetent cells of *L. monocytogenes* then plated on BHI$^{kan}$ agar plates and incubated at 30°C to select transformants, as previously described [34]. To generate the *rsbX* K77N mutation, the flanking regions of *rsbX* K77 were amplified with several synonymous mutations (with similar codon frequencies) along with the

**Table 2. Primers used in this study.**

| Primers | Sequence | Primers | Sequence |
|---|---|---|---|
| *RT-qPCR* | | | |
| 16S F | TGGGGAGCAAACAGGATTAG | gadD3 F | CTGAGGAAGAAAGCACGAGT |
| 16S R | TAAGGTTCTTCGCGTTGCTT | gadD3 R | TTTTTCTCGAGCGTTTCTGC |
| aguA1 F | CAGCGATTTCCCGTTTTGAA | lmo2230 F | TGGGCGAAAAGACTTTCACT |
| aguA1 R | ACAAACCATCGACCAATCCA | lmo2230 R | TGGAAATTTTGGTGCAGTTTCA |
| gadT2 F | ATCCAACATTTGCCACTTCC | inlA F | CGATTAGTGATATTAGTGCGCTTTC |
| gadT2 R | AAGAAGATTGCGGCAAAACC | inlA R | GGTTGTTAGTAGCGATAAGACTTTC |
| *Cloning* | | | |
| rsbX Up F | ATATGGATCCTCTGTAGATCATTCGATTGA | | |
| rsbX Up R | GCTTGGTTGGCTTTTTCTAACATATCGGTAATATCTGCAT | | |
| rsbX Down F | TGTTAGAAAAAGCCAACCAAGCTGTTTCAGGACTTCGAGG | | |
| rsbX Down R | ATATGTCGACAAGATTGCTTCACGCTTATT | | |
| rsbW ATG eGFP F | ATATCCATGGAAATCAAAAATGTAGAGGGTGAAATGAATGGCAACAATGCATGACATGGTTAG-TAAAGGAGAGGA | | |
| rsbW ATA eGFP F | ATATCCATGGAAATCAAAAATGTAGAGGGTGAAATGAATTGCAACAATGCATGACATGGTTAG-TAAAGGAGAGGA | | |
| rsbW ATT eGFP F | ATATCCATGGAAATCAAAAATGTAGAGGGTGAAATGAATAGCAACAATGCATGACATGGTTAG-TAAAGGAGAGGA | | |
| plcB ATG eGFP F | ATATCCATGGGAAAAAATAGTTGAAGACGCAGAATGAAAGAAAAAGTGAGGTGAGTGATAT-GAAGTTCAAAAAAGTGATGGTTAGTAAAGGAGAGGA | | |
| plcB GTG eGFP F | ATATCCATGGGAAAAAATAGTTGAAGACGCAGAATGAAAGAAAAAGTGAGGTGAGTGATGT-GAAGTTCAAAAAAGTGATGGTTAGTAAAGGAGAGGA | | |
| lftR ATG eGFP F | ATATCCATGGTACTTTTTCAATAGTAGTAACAACTAGTAGAGGAGTGAATTCAAATGAAAG-GACTAACCGAGTTACTCATGGTTAGTAAAGGAGAGGA | | |
| lftR GTG eGFP F | ATATCCATGGTACTTTTTCAATAGTAGTAACAACTAGTAGAGGAGTGAATTCAAGTGAAAG-GACTAACCGAGTTACTCATGGTTAGTAAAGGAGAGGA | | |
| pIMK3 eGFP R | ATATGGATCCTTATTTGTATAATTCA | | |

K77N mutation. The two flanking regions were fused by SOEing PCR and ligated to pMAD to create pMAD::*rsbX* K77N [74]. This vector was transformed to MQ140025 strain by electroporation and the rest of mutagenesis procedure was carried out according to the previously described protocol [28].

## Transcriptional analysis

Methods for transcriptional analysis were adapted from Wu et al., (2023). Briefly, 1 mL of stationary culture was mixed with 5 mL RNAlater (Sigma) and incubated for 5 min at room temperature. Then cells were resuspended (8,000 rpm × 5 min, 4°C) using 700 μL cold buffer RLT (Qiagen RNeasy kit) in lysis matrix B tubes, for mechanical lysis (FastPrep, 40 s × 6 m s$^{-1}$, twice). The rest of the protocol was carried out following the manufacturer's instructions (Qiagen RNeasy kit). DNA contamination was first depleted from the resulting RNA preparation using TURBO DNase (Invitrogen), then reverse transcription and qPCR analyses were carried out as previously described [28]. Relative gene expression was calculated using 16S as a reference gene [75].

## Fluorescence quantification

Fluorescence levels from reporter strains were used as a measure of eGFP synthesis. Reporter strains were grown in BHI$^{kan}$ at 37°C for 18 h with or without 1 mM IPTG, then 1 mL culture was washed twice using 700 μL PBS with

chloramphenicol (5 µg/mL). For each sample, 100 µL cell suspension was loaded to a 96 wells plate in duplicates. The fluorescence (489nm/518nm) and OD600nm at 20°C for each sample was measured using a microplate reader (Synergy H1, Agilent) with PBS containing wells used as blanks. Results were calculated and presented as 500 x (RFU/OD). Three independent experiments were performed.

### Immunoblotting method

For crude protein extraction, cells were collected from 5 mL of 18 h overnight culture by centrifugation (8,000 rpm × 5 min). Chloramphenicol (10 µg/mL) was added at the point of harvesting to disable protein synthesis during the protein extraction procedure. Cell pellets were resuspended in 2 mL sonication buffer (13 mM Tris-HCl, 0.123 mM EDTA, and 10.67 mM MgCl$_2$, pH 8.0) with lysozyme (1 mg/mL) and incubated at 37°C for 30 min. Then cells were resuspended in 200 µL ice-cold sonication buffer with protease inhibitor (cOmplete, Roche) and transferred to 2 mL tubes with 0.25 mL of 0.5 mm zirconia beads and 0.5 mL of 0.25 mm zirconia beads (Thistle Scientific) for mechanical lysis (FastPrep 6 m/s × 40 s, three times). Cell lysate was centrifuged at 4°C at 13,000 rpm for 30 min, and supernatant was collected. Protein samples were collected from three independent experiments.

For immunoblotting, protein concentration was determined using the BCA assay (Pierce BCA Protein Assay Kits, ThermoFisher) according to manufacturer's instruction. For each sample, 10 ng protein was separated in SDS-PAGE (12% acrylamide-bisacrylamide) and transferred to polyvinylidene difluoride membrane (PVDF). PVDFG membranes were blocked in Tris-buffered saline with 0.1% tween (TBST) with 5% (w/v) skim milk powder at room temperature for 30 min. Primary antibody (rabbit anti-GFP, polyclonal, Invitrogen) was diluted 4000 × in TBST with 5% milk and incubated with membranes at 4°C overnight. The membranes were washed three times (TBST, 5 min at room temperature) before 1 h of incubation with secondary antibody at room temperature (mouse anti-rabbit, Santa Cruz, 5000 × diluted in TBST). After three additional washes (TBST, 5 min at room temperature), the membranes were incubated with chemiluminescence detection reagents (Amersham) for 5 min in the dark and visualized in a LI-COR C-DiGit chemiluminescence channel with "high sensitivity".

### Start codon usage analysis

A local blast DNA database was created for each *L. monocytogenes* genome sequence that was deposited at NCBI database (n = 60,690; access date: 24th May 2024). Fasta format protein sequences of *L. monocytogenes* strain EGD-e were extracted from Listiwiki [76] to use as a query (each protein sequence was used as query in tblastn to search for matching DNA sequence in each DNA database). The "- seg" option was enabled for tblastn to obtain full length genes that have N-terminal amino acid sequences of low complexity. The subject sequence name and coordinate of each top tblastn hits was recorded, and subsequently used for extracting DNA sequence from fasta format genome sequences. A gene is considered present in a genome if tblastn retrieves full length protein sequence from the DNA database of this genome. The first three nucleotides in each extracted DNA sequence were recorded as the start codon. For each genome, *in silico* multi-locus sequence typing was performed using mlst [72]. CC and lineage was assigned by cross checking sequence types to the MLST typing scheme: "listeria_2". Genes were defined as conserved if they were present with a complete ORF in >95% (>57,655) of genomes. Those genes reported as showing flexible SC usage were defined as genes where the SC was different in more than 5% of the sequences analysed. The full dataset including infrequent changes (<5%) are presented in S1 Table.

## Supporting information

**S1 Table. Start codon usage of 2265 conserved genes among 60690 genomes of *L. monocytogenes*.**
(XLSX)

**S1 Fig. Phylogenetic inference of MQ140025.** (A) Core-genome based phylogeny of ST1 5446 genomes. Genetic clade in which MQ140025 was placed is highlighted in colour. (B) Detailed illustration of MQ140025 and its close relatives within coloured branch in panel A.
(DOCX)

**S2 Fig. The growth performance and SigB-dependent genes expression of MQ140025 strains.** Growth curves at 37°C are shown (A). Transcripts levels of *gadD3* (B), *lmo2230* (C) and *rsbW* (D) at stationary phase were measured and expressed relative to MQ140025 strain with wild type *sigB* operon using 16S as reference gene. Transcripts levels of *sigB* at stationary phase were measured and expressed relative to MQ140025 strain with wild type *sigB* operon using *rsbW* as reference gene (E). For transcriptional analysis, three independent experiments were carried out and each with technical duplicates. All statistical significance was calculated between each strain to MQ140025 using paired two-tailed *t*-test (ns, not significant; *, $P<0.05$; **, $P<0.01$; and ***, $P<0.001$).
(DOCX)

**S3 Fig. SC are selected during evolutionary adaption to stress and pathogenesis.** The SC usage in several clinical- or food- prevalent CC of *L. monocytogenes* are presented for 39 genes showing evidence of alternative SC usage, with lineage I and lineage II CCs in pink and blue backgrounds, respectively.
(DOCX)

**S1 Data. Including all raw experimental data underlying this study.**
(ZIP)

**S2 Data. Relevant data.** Start codon analysis output for all coding sequence analysed from a total of 60,690 *L. monocytogenes* genome sequences.Each file is tabular format output generated with tblastn, mlst, and start codon analysis.The columns are in such order:"tblastn subject sequence id", "tblastn subject start", "tblastn subject end", "tblastn subjectlength", "tblastn percentage iden:ty", "tblastn query length", "tblastn length", "Genome filename", "Start codon usage".
(PDF)

## Acknowledgments

We are grateful to all the members in Bacteria Stress Response Group for fruitful discussions. We thank Dr. Kate Reddington, University of Galway, for help with Nanopore sequencing data.

## Author contributions

**Conceptualization:** Jialun Wu, Duarte N. Guerreiro, Conor O'Byrne.

**Data curation:** Jialun Wu.

**Formal analysis:** Jialun Wu.

**Funding acquisition:** Catherine M. Burgess, Conor O'Byrne.

**Investigation:** Jialun Wu, Claire Kelly, Brenda Chanza, Ashley Reade, Catherine M. Burgess, Conor O'Byrne.

**Methodology:** Jialun Wu, Claire Kelly, Duarte N. Guerreiro, Conor O'Byrne.

**Project administration:** Catherine M. Burgess, Conor O'Byrne.

**Resources:** Jialun Wu, Claire Kelly, Catherine M. Burgess, Conor O'Byrne.

**Software:** Jialun Wu, Conor O'Byrne.

**Supervision:** Catherine M. Burgess, Conor O'Byrne.

**Validation:** Jialun Wu, Conor O'Byrne.

**Visualization:** Jialun Wu, Conor O'Byrne.

**Writing – original draft:** Jialun Wu, Conor O'Byrne.

**Writing – review & editing:** Jialun Wu, Duarte N. Guerreiro, Conor O'Byrne.

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
