## [Decision Letter · Decision Letter 0]

16 Sep 2025

PGENETICS-D-25-00938

Translation control by altered start codon usage as a means of modulating the general stress response and virulence in Listeria monocytogenes.

PLOS Genetics

Dear Dr. Wu,

Thank you for submitting your manuscript to PLOS Genetics. After careful consideration, we feel that it has merit but does not fully meet PLOS Genetics's publication criteria as it currently stands. Therefore, we invite you to submit a revised version of the manuscript that addresses the points raised during the review process.

Please submit your revised manuscript within 60 days Nov 15 2025 11:59PM. If you will need more time than this to complete your revisions, please reply to this message or contact the journal office at plosgenetics@plos.org. Please include the following items when submitting your revised manuscript:

We look forward to receiving your revised manuscript.

Kind regards,

Danielle A. Garsin

Section Editor

PLOS Genetics

Danielle Garsin

Section Editor

PLOS Genetics

Aimée Dudley

Editor-in-Chief

PLOS Genetics

Anne Goriely

Editor-in-Chief

PLOS Genetics

**Additional Editor Comments:**

Reviewer #1:

Reviewer #2:

This reviewer mentioned several additional experiments that would strengthen and validate the findings. The one I would strongly consider is comment #4: "To validate their results, the authors need to reconstruct the rsbW start codon mutations in the parental background. Ideally, these mutations also should be transplanted into a SigB+ reference strain to determine their effect on SigB activation in a SigB+ background."

**Journal Requirements:**

2) We noticed that you used the phrase 'data not shown' in the manuscript. We do not allow these references, as the PLOS data access policy requires that all data be either published with the manuscript or made available in a publicly accessible database. Please amend the supplementary material to include the referenced data or remove the references.

- TM on line 527.

5) We notice that your supplementary Figures are included in the manuscript file. Please remove them and upload them with the file type 'Supporting Information'. Please ensure that each Supporting Information file has a legend listed in the manuscript after the references list.

Potential Copyright Issues:

i) Figure 1A. Please confirm whether you drew the images / clip-art within the figure panels by hand. If you did not draw the images, please provide (a) a link to the source of the images or icons and their license / terms of use; or (b) written permission from the copyright holder to publish the images or icons under our CC BY 4.0 license. Alternatively, you may replace the images with open source alternatives. See these open source resources you may use to replace images / clip-art:

7) In the online submission form, you indicated that All data can be made available upon request.. All PLOS journals now require all data underlying the findings described in their manuscript to be freely available to other researchers, either

1. In a public repository

2. Within the manuscript itself

3. Uploaded as supplementary information.

8) Please amend your detailed Financial Disclosure statement. This is published with the article. It must therefore be completed in full sentences and contain the exact wording you wish to be published.

2) If any authors received a salary from any of your funders, please state which authors and which funders..

9) Please send a completed 'Competing Interests' statement, including any COIs declared by your co-authors. If you have no competing interests to declare, please state "The authors have declared that no competing interests exist". Otherwise please declare all competing interests beginning with the statement "I have read the journal's policy and the authors of this manuscript have the following competing interests"

**Reviewers' comments:**

Reviewer's Responses to Questions

**Comments to the Authors:**

Reviewer #1: In the research article titled “Translation control by altered start codon usage as a means of modulating the general stress response and virulence in Listeria monocytogenes.”, the authors have employed an in vitro evolution experiment to genetically dissect the ability of the reference strain to adapt to enhanced acid stress. They concluded that the acid-resistant phenotypes of the reference strain variants were due to mutationally acquired start codon variations in the SigB regulon, specifically in the rsbW gene, which allowed for translational control of RsbW expression and thereby modulation of SigB activity. Additionally, the authors analyzed all publicly available genomes of Listeria monocytogenes strains to gauge better the differences in virulence and stress response gene expression levels resulting from flexible start codon usage.

Overall, the study is well-designed, and the hypothesis is well-tested. However, I had some comments and suggestions for the authors that would further improve the rigor of this study and the representation of the data:

• Fig. 1A, B, and the text in lines 128-139 describing the results are unclear to me. Figure 1A shows that the cultures were subjected to daily cycles of stress at either pH 3 or pH 2.5 for 4 days. However, in Figure 1B, it seems that the pH 3 stress was given for the first two days, followed by pH 2.5 stress for days 3 and 4. Could the authors please provide more clarification as to which of the two approaches was used? Authors also need to mention in the results that the duration of acid stress was steadily increased on subsequent days for recoverable samples.

• Have the authors measured the transcript levels of sigB in the ARDs? It would be helpful to include this data as an additional panel in Fig. 2, which would further support the notion that the observed phenotypes resulted from increased activity rather than elevated levels of SigB.

• In Fig. 4C, the – and + IPTG labels appear to be swapped. Please correct this. The authors also need to include a control showing that SigB levels were not affected -/+ IPTG. Alternatively, the authors may include a transcriptional reporter plot for sigB in the ARDs to illustrate this point.

• As part of the Fig. 4D illustration, it would be helpful to readers if the authors included more details of the SigB regulon (for example, the binding of SigB to the P2 promoter of prfA), as this provides context for subsequent analyses.

• Information about the statistical analyses performed is missing in the figure legends of both Fig. 4 and 5. Please include these.

• Minor:

o Line 51: Remove “available”.

o Line 152: Typo “permanent”

o Line 344: Typo “ancestral”

o Line 831-832: three “parallel cultures”….were “repeatedly” exposed…

Reviewer #2: The manuscript by Wu et al. describes an interesting observation regarding the occurrence of adaptive mutations in the regulatory circuit controlling activity of the alternative sigma factor SigB in the bacterium Listeria monocytogenes. SigB is a critical regulator of the virulence of this pathogen and mutations in the genes controlling its activity have recently been shown to explain differences between hypo- and hypervirulent sublineages (PMID: 39578578). Thus, the authors address a significant topic.

The paper mainly describes suppressor mutants that arose when a clinical strain with lowered SigB activity (due to mutations in the rsbU and rsbX genes of the sigB operon) is challenged with acid stress, for which the SigB response is required to handle this. These suppressors were more resistant to acid stress, both at pH 3 and pH 2.5, and some of them formed smaller colonies on agar plates and showed low motility, resembling known small colony variants with increased SigB activity. To test the hypothesis that SigB activity is restored in these suppressors, the authors quantified transcription of known SigB-dependent genes. These genes (inlA, lmo2230) were in fact upregulated in most suppressors, whereas upregulation of SigB-independent acid stress response genes was not found to be upregulated (agu). Genome sequencing of the SigB-deficient clinical isolate and several acid resistant suppressors confirmed the rsbU *317Q mutation in the parental strain and identified rsbW mutations in the acid resistant suppressors. The mutations found in rsbW were either nonsense mutations or mutations affecting the rsbW start codon (which at the same time affect the C-terminus of the upstream gene rsbV). To test the effect of the start codon mutations found on translation efficiency of rsbW, a reporter was constructed that contained the N-terminus of RsbW fused to eGFP. Using this reporter, the authors show that ATG is more efficiently used as a start codon than ATA (~30% of ATG) and ATT (~0%) explaining the stimulating effect of these mutations on SigB activity. Analysis of start codon usage in the sigB operon genes among 70.000 genomes showed that non-canonical start codons are extremely rare and therefore unlikely to explain virulence differences between different phylogenetic sublineages (which are more common). In a more systematic analysis of start codon usage of 2180 core genome genes, non-canonical start codons were observed for 42 core genome genes, among them plcB (phospholipase C) and lftR (transcriptional regulator), for which the effect of an ATG->GTG exchange was quantified in a reporter assay.

Although the study is interesting and reports on a previously overlooked aspect that may be relevant to the virulence of L. monocytogenes, I consider it to be a preliminary study that needs to be complemented by additional experiments to further demonstrate the relevance of its main findings.

General concerns:

1. The parental strain used here has lowered SigB-activity due to two rsb mutations. The relevance of this genetic constellation is unknown. How frequent are these mutations among clinical and/or environmental strains?

2. How relevant is this phenomenon for SigB+ strains? Does a similar phenomenon (occurrence of non-canonical start codons in rsbW) can be observed in wild type strains during stress exposure to constitutively switch on SigB?

3. The study is focused on the effect of the rsbW start codon mutations on acid stress survival and expression of few sigB-dependent genes. To make their results more relevant, the authors should demonstrate that non-canonical start codons in rsbW increase virulence.

4. To validate their results, the authors need to reconstruct the rsbW start codon mutations in the parental background. Ideally, these mutations also should be transplanted into a SigB+ reference strain to determine their effect on SigB activation in a SigB+ background.

5. The section on the occurrence of alternative start codons is interesting but it would gain additional importance if the effect of selected start codon changes in the most interesting genes on virulence would be determined.

Specific remarks:

Line 177: The authors conclude here that only gadD3 was upregulated in the “LM” group, but at least two out of the LM strains show upregulation of gadT2, too, and gadD3 was also increased in ARD6. This is irritating and needs clarification.

Line 178: The section on ArgR needs rephrasing. It is not clear which changes in the ArgR amount are meant? Changes that could have been deduced from the occurrence of deviations in the aguA1 transcript levels?

Fig. 2A-B: More contrast is needed.

Line 193: The genome of the MQ strain was re-constructed using long read sequencing data only? Long read data need to be polished by short read data (Illumina) to remove SNPs which are introduced with high frequency in nanopore sequencing data.

Line 204: Why did the resequencing of the MQ strain not confirm the rsbX N77K mutation mentioned in the introduction?

Line 216: An explanation is needed that rsbV and rsbW overlap.

Line 239: “was” is missing

Table S1: Please include gene names where available. Why is lmo0205 (plcB) or lftR not included in this table? Do they not belong to the conserved core genes? It would be helpful to include data on the frequency of alternative start codons for the 39+2 genes that are explicitly discussed in the text here as well.

**Have all data underlying the figures and results presented in the manuscript been provided?**

Large-scale datasets should be made available via a public repository as described in the *PLOS Genetics*
data availability policy, and numerical data that underlies graphs or summary statistics should be provided in spreadsheet form as supporting information., and numerical data that underlies graphs or summary statistics should be provided in spreadsheet form as supporting information., and numerical data that underlies graphs or summary statistics should be provided in spreadsheet form as supporting information., and numerical data that underlies graphs or summary statistics should be provided in spreadsheet form as supporting information.

Reviewer #1: Yes

Reviewer #2: None

PLOS authors have the option to publish the peer review history of their article (what does this mean?). If published, this will include your full peer review and any attached files.). If published, this will include your full peer review and any attached files.). If published, this will include your full peer review and any attached files.). If published, this will include your full peer review and any attached files.

...

Reviewer #1: No

Reviewer #2: No

**Figure resubmission:**
---

## [Decision Letter · Decision Letter 1]

10 Mar 2026

PGENETICS-D-25-00938R1

Translation control by altered start codon usage as a means of modulating the general stress response and virulence in Listeria monocytogenes.

PLOS Genetics

Dear Dr. Wu,

Thank you for submitting your manuscript to PLOS Genetics. After careful consideration, we feel that it has merit but does not fully meet PLOS Genetics's publication criteria as it currently stands. Therefore, we invite you to submit a revised version of the manuscript that addresses the points raised during the review process.

Please submit your revised manuscript within by Apr 09 2026 11:59PM. If you will need more time than this to complete your revisions, please reply to this message or contact the journal office at plosgenetics@plos.org. Please include the following items when submitting your revised manuscript:

We look forward to receiving your revised manuscript.

Kind regards,

Danielle A. Garsin

Section Editor

PLOS Genetics

Danielle Garsin

Section Editor

PLOS Genetics

Aimée Dudley

Editor-in-Chief

PLOS Genetics

Anne Goriely

Editor-in-Chief

PLOS Genetics

**Additional Editor Comments:**

The reviewers are largely satisfied with the revisions and offer just a few minor corrections. Reviewer #2 is still concerned about not being able to introduce the start site mutation into a clean background. However, I am satisfied with the authors' explanation for why this proved impossible and the additional information provided to address this point.

**Reviewers' comments:**

Reviewer's Responses to Questions

**Comments to the Authors:**

Reviewer #1: The authors have largely addressed the reviewers' concerns and suggestions in their revised manuscript. I therefore recommend publication of this article.

Minor edits:

Line 247-248: Figure 1A needs to be changed to Figure 4A.

Line 261: Figure 1A needs to be Figure 4A.

Reviewer #2: The authors have satisfactorily answered most of the points of criticism I raised. However, they leave open the question of whether the generation of rsbW start codon mutations in the POI parental strain using homologous recombination techniques is sufficient to cause the same phenotypic consequences as in the ARD suppressors. Even if technical limitations make such experiments impossible in their case, I consider this proof to be essential, since mutations can always remain undetected in genome sequencing of suppressor strains, especially duplications, rearrangements, and unstable mutations. Therefore, an important argument needed to support their hypothesis remains missing in my eyes. I leave it to the editor to decide whether an experiment of this kind needs to be included in order to meet the standards of PLOS Genetics.

Special comments

- Fig. 4D: The legend indicates that the line for the POI strain contains green circles, which, however, cannot be found in the diagram.

**Have all data underlying the figures and results presented in the manuscript been provided?**

Large-scale datasets should be made available via a public repository as described in the *PLOS Genetics*
data availability policy, and numerical data that underlies graphs or summary statistics should be provided in spreadsheet form as supporting information., and numerical data that underlies graphs or summary statistics should be provided in spreadsheet form as supporting information., and numerical data that underlies graphs or summary statistics should be provided in spreadsheet form as supporting information., and numerical data that underlies graphs or summary statistics should be provided in spreadsheet form as supporting information.

Reviewer #1: Yes

Reviewer #2: Yes

PLOS authors have the option to publish the peer review history of their article (what does this mean?). If published, this will include your full peer review and any attached files.). If published, this will include your full peer review and any attached files.). If published, this will include your full peer review and any attached files.). If published, this will include your full peer review and any attached files.

...

Reviewer #1: No

Reviewer #2: No

**Figure resubmission:**
---

## [Editor Report · Decision Letter 2]

24 Mar 2026

Dear Dr Wu,

We are pleased to inform you that your manuscript entitled "Translation control by altered start codon usage as a means of modulating the general stress response and virulence in Listeria monocytogenes." has been editorially accepted for publication in PLOS Genetics. Congratulations!

Yours sincerely,

Danielle A. Garsin

Section Editor

PLOS Genetics

Danielle Garsin

Section Editor

PLOS Genetics

Aimée Dudley

Editor-in-Chief

PLOS Genetics

Anne Goriely

Editor-in-Chief

PLOS Genetics

BlueSky: @plos.bsky.social

Comments from the reviewers (if applicable):

**Data Deposition**

If you have submitted a Research Article or Front Matter that has associated data that are not suitable for deposition in a subject-specific public repository (such as GenBank or ArrayExpress), one way to make that data available is to deposit it in the Dryad Digital Repository. As you may recall, we ask all authors to agree to make data available; this is one way to achieve that. A full list of recommended repositories can be found on our . As you may recall, we ask all authors to agree to make data available; this is one way to achieve that. A full list of recommended repositories can be found on our . As you may recall, we ask all authors to agree to make data available; this is one way to achieve that. A full list of recommended repositories can be found on our . As you may recall, we ask all authors to agree to make data available; this is one way to achieve that. A full list of recommended repositories can be found on our website....

http://datadryad.org/submit?journalID=pgenetics&manu=PGENETICS-D-25-00938R2

Additionally, please be aware that our data availability policy requires that all numerical data underlying display items are included with the submission, and you will need to provide this before we can formally accept your manuscript, if not already present. requires that all numerical data underlying display items are included with the submission, and you will need to provide this before we can formally accept your manuscript, if not already present. requires that all numerical data underlying display items are included with the submission, and you will need to provide this before we can formally accept your manuscript, if not already present. requires that all numerical data underlying display items are included with the submission, and you will need to provide this before we can formally accept your manuscript, if not already present.

**Press Queries**

If you or your institution will be preparing press materials for this manuscript, or if you need to know your paper's publication date for media purposes, please inform the journal staff as soon as possible so that your submission can be scheduled accordingly. Your manuscript will remain under a strict press embargo until the publication date and time. This means an early version of your manuscript will not be published ahead of your final version. PLOS Genetics may also choose to issue a press release for your article. If there's anything the journal should know or you'd like more information, please get in touch via plosgenetics@plos.org....

---

## [Editor Report · Acceptance letter]

PGENETICS-D-25-00938R2

Translation control by altered start codon usage as a means of modulating the general stress response and virulence in Listeria monocytogenes.

Dear Dr Wu,

We are pleased to inform you that your manuscript entitled "Translation control by altered start codon usage as a means of modulating the general stress response and virulence in Listeria monocytogenes." has been formally accepted for publication in PLOS Genetics! Your manuscript is now with our production department and you will be notified of the publication date in due course.

With kind regards,

Anita Estes

PLOS Genetics

On behalf of:
